# The automatic parameter-exploration with a machine-learning-like approach: Powering the evolutionary modeling on the origin of life

**Yuzhen Liang**[1], **Chunwu Yu**[2], **Wentao Ma**[1]*

**1** Hubei Key Laboratory of Cell Homeostasis, College of Life Sciences, Wuhan University, Wuhan, China,
**2** College of Computer Sciences, Wuhan University, Wuhan, China

* mwt@whu.edu.cn

## Abstract

The origin of life involved complicated evolutionary processes. Computer modeling is a promising way to reveal relevant mechanisms. However, due to the limitation of our knowledge on prebiotic chemistry, it is usually difficult to justify parameter-setting for the modeling. Thus, typically, the studies were conducted in a reverse way: the parameter-space was explored to find those parameter values "supporting" a hypothetical scene (that is, leaving the parameter-justification a later job when sufficient knowledge is available). Exploring the parameter-space manually is an arduous job (especially when the modeling becomes complicated) and additionally, difficult to characterize as regular "Methods" in a paper. Here we show that a machine-learning-like approach may be adopted, automatically optimizing the parameters. With this efficient parameter-exploring approach, the evolutionary modeling on the origin of life would become much more powerful. In particular, based on this, it is expected that more near-reality (complex) models could be introduced, and thereby theoretical research would be more tightly associated with experimental investigation in this field–hopefully leading to significant steps forward in respect to our understanding on the origin of life.

## Author summary

People have long been interested in the evolutionary processes through which life on our planet could have arisen from a non-life background. However, it seems that experimental studies in this field are proceeding slowly, perhaps owing to the complication of such processes. In the meantime, computer modeling has shown its potential to disclose the evolutionary mechanisms involved. Now a major difficulty of the computer modeling work is to justify the parameter-setting–on account of our limited knowledge on prebiotic chemistry and environments. Thus, people tend to explore the parameter space to seek parameter values in favor of the hypothetic scene and leave the parameter-justification a later job when sufficient knowledge is available. To date, the parameter-exploration is usually conducted manually (in many cases by trial and error), thus arduous and unpredictable. Inspired by the algorithm of machine-learning, we designed an automatic approach of

**Data Availability Statement:** All relevant data are within the manuscript and its Supporting Information. The source code of the approach can be obtained from: https://github.com/mwt2001gh/

automatic-parameter-exploration-in-modeling-the-origin-of-life/blob/main/mlp-e%3D0.2.cpp. The version corresponds the red-line case in Fig 2A.

**Funding:** This study was supported by the National Natural Science Foundation of China (No. 31571367)(http://www.nsfc.gov.cn) to WTM and Natural Science Foundation of Hubei Province (CN) (No.2019CFB685)(http://kjt.hubei.gov.cn) to WTM. The funders had no role in study design, data collection and analysis, decision to publish, or preparation of the manuscript.

**Competing interests:** The authors have declared that no competing interests exist.

parameter-exploration. The results showed that the approach is quite effective–that is, "good" parameter-sets in favor of hypothetic scenes in the origin of life can be found automatically. It is expected that such a machine-learning-like method would greatly enhance the efficiency of our evolutionary modeling studies on the origin of life in future.

## Introduction

Starting from a prebiotic chemical world, the scenario concerning the origin of life should have included a series of complicated processes with the combination of chemistry and evolution [1–4]. Since the pioneer work of Miller and Urey in 1950s [5], experimental studies have provided quite a lot of insights into the chemical aspect of these processes [6–8], however, the involved evolutionary processes, perhaps having lasted quite a long time (e.g., years, tens of years, or even much longer), are difficult to mimic or study in the lab. Fortunately, theoretical modeling has proven a valid way to deal with the evolutionary aspect, as exemplified by Eigen's famous work concerning "hypercycle" [9].

### Computer modeling on evolutionary processes in the origin of life

In early years, theoretical modeling in the field of the origin of life was typically based upon chemical reaction kinetics, which derived a group of differential equations to describe the dynamics of the target system–thus the so-called "differential equation model". Most parameters for the model are simply rates of various reactions. The differential equation group is usually too complicated to obtain an analytical solution, and therefore, primordial researches along this line usually only analyzed the equilibrium point(s) of the model system [10–12]. By such modeling, the knowledge we obtained about the relevant evolutionary process was quite limited.

Fortunately, we may also get aware of the evolution mechanism if we can "see" the behavior of the model system under different conditions–that is, the theoretical mimicking or simulation. For the differential equation model, we can obtain its "numerical solution" through "numerical integration" and thereby "observe" the model system's behavior. Aided by computer technology, numerical solution of complex differential equation models became a conventional approach in this area, which just represented early work of "computer simulation" (or termed "computer modeling") in the origin of life [13–18]. Later on, another sort of computer simulation also arose, i.e. the so-called "Monte-Carlo simulation". Wherein, a probabilistic model is established and the model system "runs" through sampling on random numbers iteratively–results are obtained by statistics on certain key variable(s), which reflect the behavior of the system. To date, Monte-Carlo simulation has become the most important method in the theoretical investigation on the evolutionary aspect of the origin of life [19–29] (here the reference list is certainly not a complete one).

### The significance of parameter-space exploration

A normal way for computer modeling is to set the parameters in the model according to our knowledge on reality, and run the model (conduct "simulation") to see the outcome–then, we may make prediction for the behavior of the target system. However, owing to our limitation of knowledge concerning prebiotic environments and chemistry, it is usually difficult to justify the value-setting of the parameters used in relevant modeling studies. For example, the Miller-Urey experiment assumed a reducing atmosphere for the prebiotic earth [5]; however, this

putative scene was later doubted in the field of geochemistry [30,31], and no clear conclusion has yet been achieved so far.

Therefore, in the field of the origin of life, computer simulation studies were usually conducted in a reverse way: the model's parameter space was explored to find out parameter values which may support a hypothetic evolutionary process. That is, we may expect to assess the "legitimacy" of the "favoring parameter values" later, when relevant chemical or environmental knowledge is available. On the other hand, if we are confident about the hypothetic process because it is "reasonable" or has "conspicuous supporting evidence" (most notably, e.g., "the RNA world" [32–35]), we may make some deduction on relevant prebiotic conditions according to the "favoring parameter values" for that process, thereby even likely improving our knowledge on prebiotic environments and chemistry.

In previous studies, people used to explore the parameter space manually (typically not presented explicitly in their papers–perhaps because it is difficult to characterize the route of the manual exploration), which is an arduous job, especially for the complicated modeling comprising many parameters (notably, the parameter space increases exponentially with parameter numbers). As a consequence, researchers in this area tend to adopt modeling systems quite abstract (thus involving fewer parameters)–some studies even resorted on oversimplified models (e.g., the so-called "toy models" or even "artificial chemistry" [36–38]), which appeared doubtful in their relevance to the reality concerning the origin of life. The awkward situation raises an urgent issue: can we do the parameter-exploration in a more automatic way?

## The application of machine-learning approaches

Machine-learning is a special kind of computer algorithms (as called, "master algorithm" [39] or the algorithm of artificial intelligence), seeking to automatically train a target model (or function) aiming at various tasks (e.g., classification, image recognition and natural language processing). In particular, a branch of machine-learning, "connectionism", uses the so-called "artificial neural network" as its target model [39], thereby mimicking the human brain from the basic structure and essential mechanisms–thus is expected to realize veritable artificial intelligence. Later on, the power of such machine-learning turned out to be associated with the 'depth' of the artificial neuron network, and the so-called "deep learning" has achieved tremendous successes in a variety of application fields [40].

In fact, the model training in machine-learning is, by and large, just to explore parameters automatically (there are often many parameters as well, e.g., in the complex artificial neuron network of "deep learning"), seeking to find "appropriate" parameter values for the model to be adept at the aimed task. Getting inspiration from this, we ask: "can we conduct our parameter-exploration automatically in a similar way?" That is, using the evolutionary model as the target model to train, can we explore the parameter-space automatically and thereby find "appropriate parameter-settings" resulting in the evolutionary behavior we suppose?

## Results

Firstly, we investigated a case derived from one of our previous modeling studies, relevant to early evolution in the scenario of the RNA world [20]. In that study, we supposed that a ribozyme that catalyzes the synthesis of nucleotides (nucleotide synthetase ribozyme, "NSR" for short) may have become thriving in a prebiotic RNA pool, because this ribozyme could synthesize nucleotides around itself, thus favoring its own replication. Therein, we explored the parameter space manually and found out "appropriate parameter values" supporting the prosperity of NSR in the model system [20] (Fig 1A shows a typical case). Now we intend to do the parameter-exploration automatically based upon the idea of machine-learning.

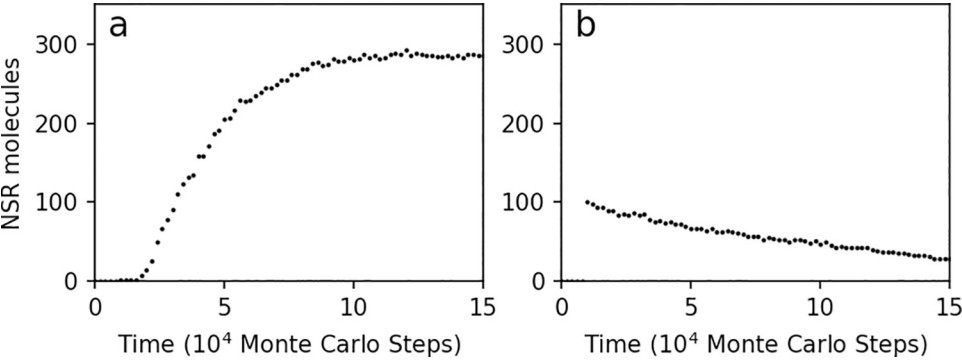

**Fig 1. The evolutionary dynamics of molecular number of NSR in the modeling.** (**a**) A typical case for the spread of NSR. One NSR molecule is inoculate at step $1 \times 10^4$, and it replicates and becomes prosperous in the system [20]. Parameter values: $PNF = 2 \times 10^{-4}$, $PNFR = 0.9$, $PND = 0.01$, $PRL = 2 \times 10^{-6}$, $PBB = 1 \times 10^{-6}$, $PAT = 0.1$, $PFP = 0.01$, and $PMV = 0.01$ (note that for a concise description we omit the other five parameters in the model which are not involved in the parameter-exploration here). (**b**) A case we assumed as the starting point for the machine-learning. One hundred NSR molecules are inoculated at step $1 \times 10^4$, and the molecular number decreases gradually. Parameter setting: $PNF = 4 \times 10^{-3}$, $PNFR = 0.02$, $PND = 1 \times 10^{-3}$, $PRL = 2 \times 10^{-5}$, $PBB = 1 \times 10^{-5}$, $PAT = 0.5$, $PFP = 0.1$, and $PMV = 1 \times 10^{-3}$. Obviously, such a parameter-setting does not favor the spread of the NSR. The NSR number at step $15 \times 10^4$ (i.e., the final step shown here) is adopted as the reference criterion–i.e. the objective function for machine-learning, which is expected to be improved through the following automatic parameter-adjustment (exploration).

There are totally thirteen parameters in the mode [20], and here we concentrate on eight of them (S1 Table) to conduct automatic exploration. In that study, we inoculated one NSR molecule to see if such molecules could spread (become thriving by replication) in the system (Fig 1A). In fact, the number of NSR molecules a definite period of time after the inoculation may serve as a target value reflecting the spread tendency of the NSR. That is to say, via automatic parameter-adjustment, this number is expected to increase–in terms of machine-learning, "the objective function". However, for the starting point of the machine-learning–assumedly with a "bad parameter set", one NSR molecule would tend to degrade by chance before it can give rise of more offspring (thus no learning feasible). Therefore, we inoculated 100 NSR molecules initially–so that we see a decline of NSR molecules rather than an immediate extinction (Fig 1B). Then, the NSR number at a reference point within the declining curve (here, step $15 \times 10^4$) was chosen and the automatic parameter-exploration was conducted to find parameter settings favoring the increase of this "objective function". The alteration of parameter settings was expected to reverse the declining tendency and ultimately favor the spread of the NSR in the system instead.

This "machine-learning" used an optimization approach of gradient ascent–the parameters changed simultaneously in line with the steepest rising direction regarding the objective function–the NSR number (see Methods). The approach turned out to be successful, and the target NSR number rises dramatically while the parameter setting is being adjusted automatically (Fig 2). The subfigures in Fig 2 correspond to four different starting parameter sets; wherein, the curves with different colors correspond to cases adopting different learning rate. Generally, the final target NSR number tends towards a similar level (around 3500 here), and a larger learning rate means a faster learning process but greater fluctuation during the learning.

Fig 3 demonstrates the details concerning the automatic adjustment of the eight parameters during the learning process. Notably, though with different starting parameter sets, the ultimate "good parameter sets" appear to be similar (see the curves with different colors). In general, by the learning, the parameters $PRL$, $PBB$, $PFP$ and $PNF$ tend towards values rather small (these results are coincident with those by manual exploration in the previous study [20]). The

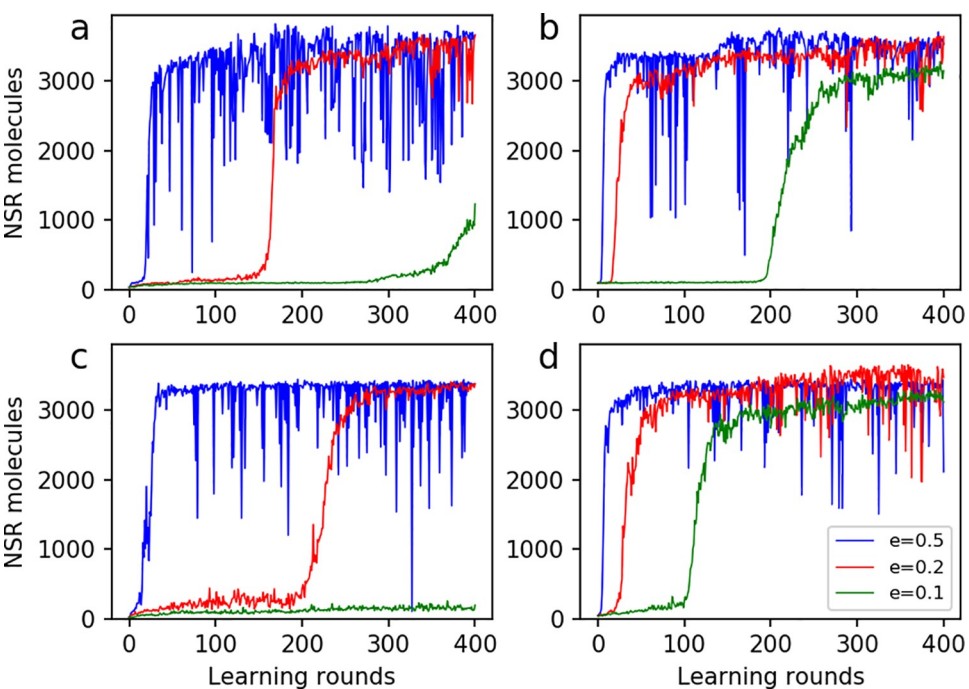

**Fig 2. The improvement of the target NSR number by the automatic parameter-exploration in principle of machine-learning.** The target NSR number refers to the number of NSR molecules at step $1.5 \times 10^5$ in the evolutionary dynamics (see Fig 1B; as the "objective function"). "e" denotes the learning rate (the color legend in the lower right subfigure applies to the whole figure). A round of learning means a round of parallel adjustment of all the eight parameters according to the greatest gradient (see Methods). The four subfigures show cases with different starting parameter values: (**a**) $PNF = 4 \times 10^{-3}$, $PNFR = 0.02$, $PND = 1 \times 10^{-3}$, $PRL = 2 \times 10^{-5}$, $PBB = 1 \times 10^{-5}$, $PAT = 0.5$, $PFP = 0.1$, and $PMV = 1 \times 10^{-3}$ (i.e., the same as those in Fig 1B); (**b**) $PNF = 4 \times 10^{-3}$, $PNFR = 0.2$, $PND = 0.01$, $PRL = 2 \times 10^{-5}$, $PBB = 1 \times 10^{-6}$, $PAT = 0.1$, $PFP = 1 \times 10^{-3}$, and $PMV = 1 \times 10^{-4}$; (**c**) $PNF = 5 \times 10^{-3}$, $PNFR = 0.2$, $PND = 0.01$, $PRL = 1 \times 10^{-5}$, $PBB = 1 \times 10^{-5}$, $PAT = 0.05$, $PFP = 0.01$, and $PMV = 0.01$; (**d**) $PNF = 1 \times 10^{-3}$, $PNFR = 0.01$, $PND = 1 \times 10^{-3}$, $PRL = 1 \times 10^{-5}$, $PBB = 1 \times 10^{-5}$, $PAT = 0.1$, $PFP = 1 \times 10^{-3}$, and $PMV = 1 \times 10^{-3}$.

results concerning other parameters (i.e., *PNFR*, *PAT*, *PMV*, *PND*) are not so straightforward, showing more complicated influence of them on the evolutionary dynamics.

Above we demonstrate the result of machine-learning for the parameter settings favoring the increase of NSR after the inoculation of 100 molecules (Fig 4A shows the greatly "improved" dynamics–in comparison with Fig 1B). Indeed, under the same parameter setting, when inoculating only one NSR molecule initially, it can also spread in the system (Fig 4B)–manifesting the success of our automatic parameter-exploration (in fact, here there are about 800 NSR molecules at the ultimate balance, apparently better than the result in our previous study–i.e. less than 300 NSR molecules in Fig 1A).

However, we noticed a detail which looks abnormal. In the previous modeling study [20], we had supposed that NSR may have spread naturally in the system because this ribozyme could accumulate building blocks of RNA around itself, thus favoring its own replication. If so, a higher catalytic rate, which corresponds to a greater *PNFR* (refer to S1 Table), should favor the spread of the NSR. However, from the automatic parameter-adjustment curves (Fig 3), we noticed that *PNFR* does not increase as expected–it even tends to decrease. Does it mean that under the 'learned' parameter setting, the catalytic function of NSR is in practice of no use, and any RNA sequence inoculated into the system could spread? This turned out to be the case–when we inoculated a non-functional RNA sequence (the control) instead of NSR into the system, it spread as well! (Fig 4C and 4D).

   

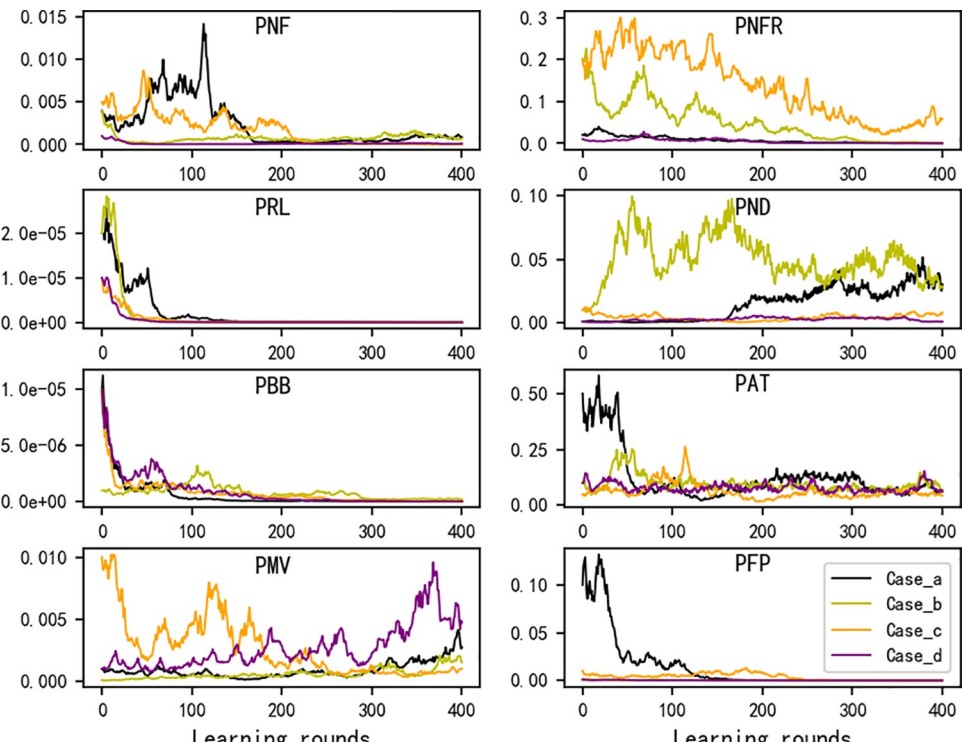

**Fig 3. The automatic parameter-adjustment during the parameter-exploration in principle of machine-learning.**
The parameter-adjustment tendency of the four cases with the learning rate of 0.2 in Fig 2A–2D (i.e., represented by
the red lines within the subfigures in Fig 2) are shown here in different colors (the legend in the lower right subfigure
applies to the whole figure). The vertical axis of a subfigure represents the value of the corresponding parameter whose
name is shown within the panel (in the *PFP* subfigure, the Case_b line is covered by the Case_d line). Note that the
eight probabilities are in practice adjusted simultaneously in a learning round (see Methods).

Interestingly, this abnormal situation offered us a chance to demonstrate the power of our
automatic parameter-exploration further. For a modification, while inoculating 100 molecules
of NSR together with 100 molecules of the control RNA species into the system, we chose the
difference between their molecular number (NSR minus Control) as the target value to be
improved in the process of machine-learning. As a result, the difference was enlarged via the
machine-learning (Fig 5A). Then, with the learned parameter set, NSR could spread in the system but the control species cannot! (Fig 6). Notably, here in the learned parameter set, *PNFR*
(the rate for enzymatic nucleotide synthesis) becomes significantly greater than *PNF* (the rate
for non-enzymatic nucleotide synthesis): *PNFR* = 0.036 and *PNF* = $8.36 \times 10^{-7}$ (see the legend
of Fig 6), in comparison with the former pair: *PNFR* = $5.51 \times 10^{-3}$ and *PNF* = $3.04 \times 10^{-4}$ (see the
legend of Fig 4). That is, the machine-learning finally achieved a result showing that NSR may
spread in the RNA pool on account of their enzymatic function–fully supporting the aforementioned hypothetic idea [20].

Doubtlessly, the power of such automatic parameter-exploration is impressive. Then it is
naturally to think that, could such an approach be bringing about an illusion? In other words,
whether the automatic exploration is "omnipotent", capable of seeking out any results one
desires, regardless of any relevant evolutionary mechanisms? In terms of machine-learning, it
is somewhat like the concept of "over-fitting". If so, the approach would be meaningless for
our goal of investigating evolutionary processes. Hence, we chose to "knock out" the function
of NSR in the modeling (i.e., the NSR no longer catalyzes nucleotide synthesis) and came back

   

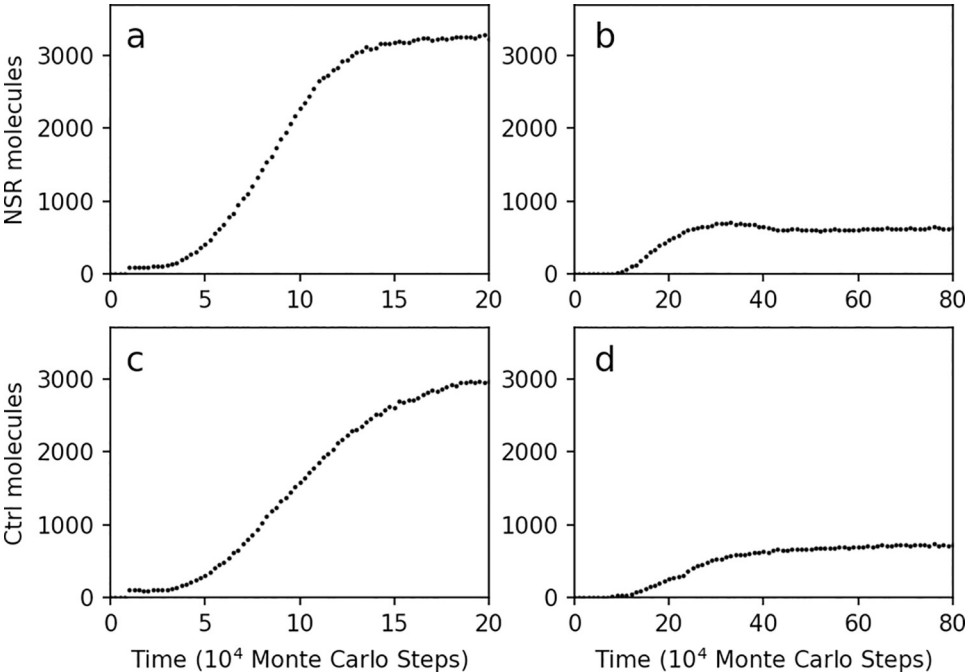

**Fig 4. The parameter setting resulting from the machine-learning turned out to favor both the spread of NSR and that of non-functional RNA species.** In reference of the case shown in Fig 2A with learning rate of 0.2 (the red line), the parameter set at the 200th round of the machine-learning is adopted: $PNF = 3.04×10^{-4}$, $PNFR = 5.51×10^{-3}$, $PND = 0.0194$, $PRL = 7.25×10^{-8}$, $PBB = 6.24×10^{-8}$, $PAT = 0.0763$, $PFP = 3.51×10^{-4}$, $PMV = 4.36×10^{-5}$. At step $1×10^{4}$, (**a**) one hundred NSR molecules are inoculated; (**b**) one NSR molecule is inoculated; (**c**) one hundred control (Ctrl) molecules, which have no enzymatic activity, are inoculated; (**d**) one control molecule is inoculated.

to the learning case aiming at enlarging the difference between the NSR and the control species. Then no difference can be "learned out"! (Fig 5B). That is to say, the aforementioned difference between the NSR and control species (Fig 5A) is indeed stemmed from the function of NSR, and our worry about the problem of "over-fitting" is not necessary.

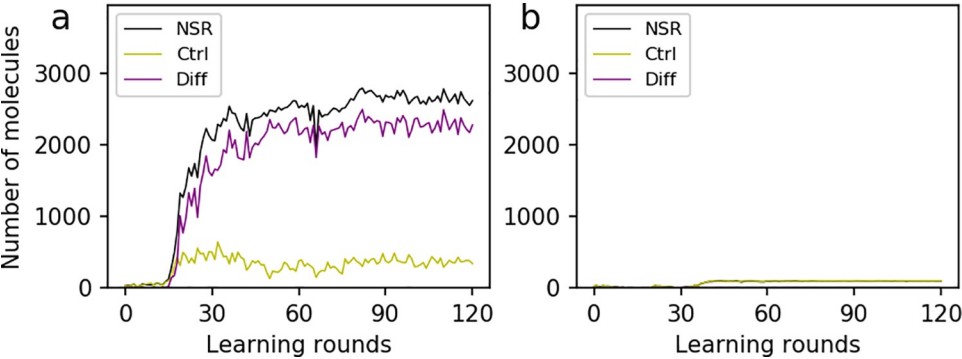

**Fig 5. Adopting the difference between NSR and the control RNA species as the objective function.** The starting value of parameters are the same as those in Fig 2A: i.e., $PNF = 0.004$, $PNFR = 0.02$, $PND = 0.001$, $PRL = 2×10^{-5}$, $PBB = 1×10^{-5}$, $PAT = 0.5$, $PFP = 0.1$, and $PMV = 0.001$. The learning rate e = 0.5. Here the objective function is the difference (Diff) between the number of NSR and the control RNA species (i.e. NSR—Ctrl), instead of the number of NSR *per se*. (**a**) NSR plays its normal role as it is named. (**b**) NSR is assumed to lose its function. In fact, in case *b*, the curve of NSR (black) is almost covered by that of Ctrl (yellowish-green) because there is nearly no difference between them throughout the whole learning process, and the curve of Diff (purple) lies very close to the horizontal axis.

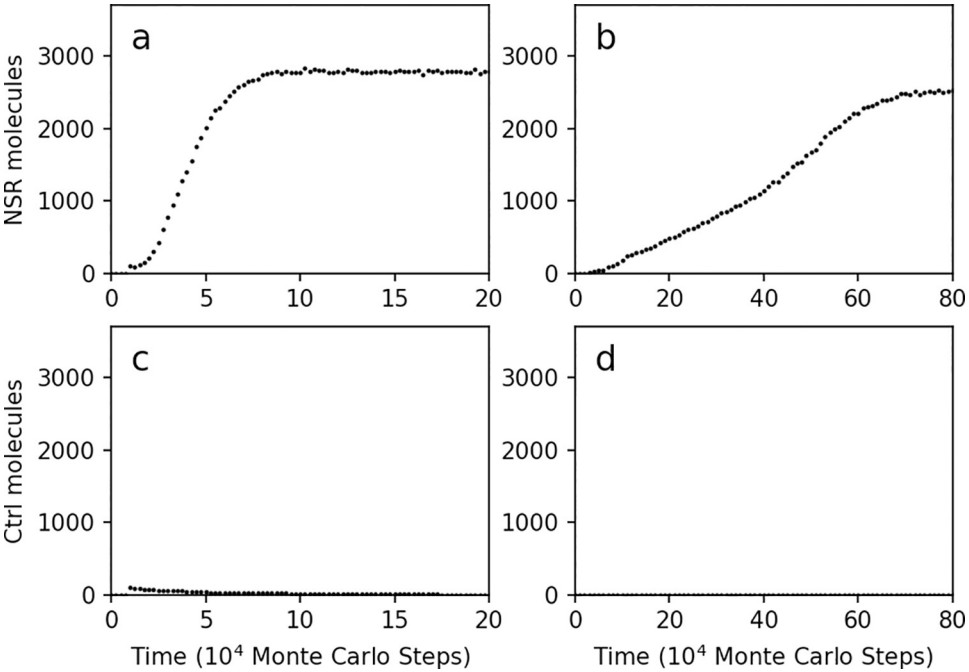

**Fig 6. Using the difference between the NSR and control molecules as the objective function resulted in the parameter set favoring the spread of NSR but not the control.** In reference of the case shown in Fig 5A, the parameter set at the 50th round of the machine-learning is adopted: $PNF = 8.36\times10^{-7}$, $PNFR = 0.036$, $PND = 3.77\times10^{-3}$, $PRL = 9.89\times10^{-8}$, $PBB = 3.05\times10^{-5}$, $PAT = 0.373$, $PFP = 2.17\times10^{-5}$, $PMV = 2.89\times10^{-4}$. At step $1\times10^4$, (**a**) one hundred NSR molecules are inoculated; (**b**) one NSR molecule is inoculated; (**c**) one hundred control (Ctrl) molecules, which have no enzymatic activity, are inoculated; (**d**) one control molecule is inoculated (the number symbols lie very close to the horizontal axis).

Above we have seen that the objective function can be modified to serve a different goal of the parameter-exploration. To a degree, this illustrates the robustness of the present method. In fact, the method is also robust even when our learning strategy is modified. For instance, in the strategy used originally, when a change of one parameter does not bring about the value change of the objective function, in the next round of learning it is adjusted upward–actually, if it is adjusted downward instead, the machine-learning method still effectively lead to the increase of the NSR (S1A Fig). Additionally, in regard of the learning rate, when we adopt a strategy of multiplication instead of a strategy of addition (see Methods for details), the machine-learning is still effective (S1B and S1C Fig). Furthermore, when we adopt the approach of coordinate ascent instead of the gradient ascent (see Methods for details), the machine-learning can also find an "appropriate" parameter setting favoring the spread of NSR (S1D Fig).

Then we are curious about whether the machine-learning method described here could apply to computer simulation studies from other groups in the field, which may just serve as a verification test of our method. A famous research of Szathmáry and coworkers [19], in which the source code of the simulation program was provided, offers us a chance to assess this concern. The work addressed an important issue concerning the plausibility of Darwinian evolution at molecular level: without membrane, could RNA-like molecules evolved towards more efficiency and greater complexity? The study assumed a relatively abstract model, in which the RNA-like polymers are referred to as "replicators". The results showed that with limited dispersal, the replicators could evolve towards higher efficiency and fidelity (in their replication). Fig 7A shows a typical case of such evolution in the computer simulation (Monte-Carlo

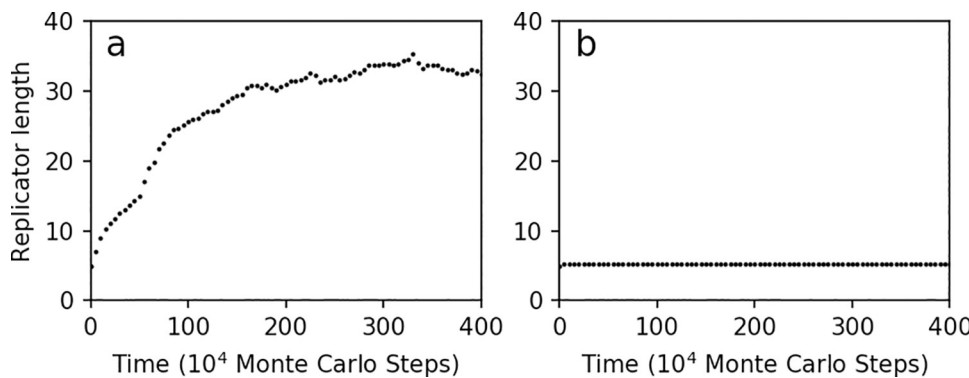

**Fig 7. The evolutionary dynamics of average length of replicators in the modeling.** (**a**) A typical case demonstrating the hypothetic scenario in the original work [19]. The average length of the replicators in the system is initially 5 (in monomers), and it increases significantly accompanying with the evolutionary process. Parameters are set according to the original work: $\alpha_A = 0.1$, $\beta_A = 3$, $\gamma_A = 200$, $\alpha_B = 0.1$, $\beta_B = 3$, $\gamma_B = 200$, $\alpha_C = 2$, $\gamma_C = 5$ (note that for clarity we omit the description of the other four parameters in the same model which are not involved in the parameter-exploration here). (**b**) A case we assumed as the starting point for the machine-learning. Parameter setting: $\alpha_A = 0.2$, $\beta_A = 2$, $\alpha_B = 0.2$, $\beta_B = 2$, $\beta_C = 1$, and the other three parameters are the same as in (a). Obviously, such a parameter-setting does not support the hypothetic idea [19]–there is nearly no increase in the average length of the replicators. The situation is expected to "improve" via the automatic parameter-exploration in principle of machine-learning that we present here.

simulation), in which the average length of the replicators in the system was monitored, marking the evolutionary process towards complexity. Obviously, here, such an average length may just serve as the objective function in our machine-learning.

To set a starting point for the machine-learning, we changed several parameter values in the case of Fig 7A, with which the average length of the replicators could not increase in the evolution (Fig 7B). The replicator length at step $100 \times 10^4$ was chosen as the objective function. Then we started the parameter-exploration in favor of the increase of the replicator length. The result of the learning manifested a clear success–the replicator length at step $100 \times 10^4$ was improved from 5 to around 30 (monomer residues) (Fig 8A), and when the learned parameter setting was applied into the evolutionary simulation, the replicator length reached about 35 at step $400 \times 10^4$ (Fig 8B)–no less than that in the typical case from the original work (Fig 7A). S2 Fig illustrates the automatic parameter-adjustment during the machine-learning.

In fact, in accordance to the case in Fig 7A, we can see that the replicator length does not reach a stable level until step $300 \sim 400 \times 10^4$ –thus a better choice for the objective function seems to be the average length of the replicators after the reach of this balance (instead of the length at step $100 \times 10^4$ as we adopted in our learning). However, if so, the computational burden would be much greater. On the other hand, we found that further intention to save the computational burden by adopting (as objective function) the replicator lengths at earlier stage may result in a limited success–e.g., the replicator length at step $20 \times 10^4$, if adopted, was merely improved to around 15 (Fig 8C), and when applying the learned parameter setting, the replicator length can only reach a level around 18 at step $400 \times 10^4$ (Fig 8D). Obviously, the parameter values optimized for the replicator length at an early stage of the evolution may not necessarily be consistent with the optimal parameter values associated with the replicator length at a later stage.

Indeed, due to complexity of the evolutionary processes in the origin of life, computational burden is always an issue deserving consideration in relevant modeling studies. Here, this issue is particularly noteworthy because perhaps many rounds of learning would be required to achieve an optimized parameter setting. Considering the "time gap" cannot be too great for an early-step target value to represent a later-step one (Fig 8), we designed a "progressive-

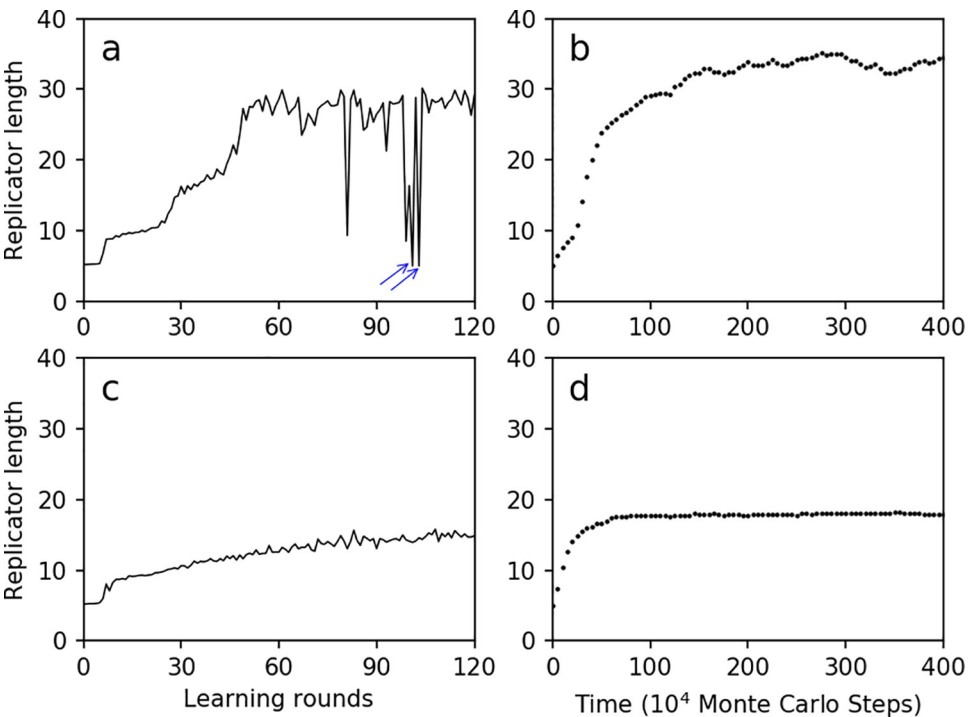

**Fig 8. The average length of replicators improves via parameter-exploration in principle of machine-learning.** (**a**) The objective function is the average length of replicators at step $100 \times 10^4$, with a starting point of learning which was assumed in case Fig 7B. The arrows indicate "odd points", which denote the rounds when the total number of replicators is zero and the average length of replicators is assumed (*ad hoc*) to adopt the starting value, i.e. 5. (**b**) The improved evolutionary dynamics (in reference of that in Fig 7B) by adopting the learned parameter set at the 60th learning round in *a*: $\alpha_A = 0.0727$, $\beta_A = 2.57$, $\gamma_A = 393$, $\alpha_B = 0.0766$, $\beta_B = 4.42$, $\gamma_B = 169$, $\beta_C = 1.88$, $\gamma_C = 3.35$. (**c**) The objective function is the average length of replicators at step $20 \times 10^4$, with a starting point of learning assumed in case Fig 7B. (**d**) The improved evolutionary dynamics (in reference of that in Fig 7B) by adopting the parameter set at the 80th learning round in *c*: $\alpha_A = 0.0358$, $\beta_A = 2.97$, $\gamma_A = 452$, $\alpha_B = 0.251$, $\beta_B = 1.44$, $\gamma_B = 143$, $\beta_C = 2.22$, $\gamma_C = 1.23$.

learning" strategy. The replicator lengths at step 10, 20, 40 and $100 \times 10^4$ are adopted as the objective functions at different stages of the machine-learning, one after another (S3A Fig). The parameter values learned from a previous stage are set as the starting point for the next learning stage. The strategy turned out to be successful–when the ultimately learned parameter setting was applied into the evolutionary simulation, the replicator length increased to a level no less than (here, even higher than) the case in which the progressive strategy is not adopted (S3B Fig). Obviously, by using the strategy, computational costs would be significantly saved while keeping the learning effective.

## Discussion

In the present study, we demonstrate that automatic parameter-exploration in principle of machine-learning may be used in the computer modeling researches adopting the reverse approach–that is, trying to find out parameter values favoring a specific outcome. As mentioned already, computer modeling researches on the evolutionary processes during the origin of life typically adopted the reverse approach. This is largely owing to the shortage of our knowledge about the prebiotic chemistry and environments, which is associated with the parameter setting. On the other hand, why this reverse approach was assumed to be particularly effective in this area? In fact, the evolution during the origin of life is remarkably characterized by the tendency from simplicity to complexity, which is a special, rare phenomenon in

nature [3,4]. Therefore, any relevant hypothetic scene in the area (suggesting a simplicity-to-complexity evolution), if supported by modeling, deserves our attention–therefore, exploring parameter-setting in favor of the scene is valuable. Or else, provided that a simplicity-to-complexity evolution were ordinary, it would make little sense to find out parameter values supporting a relevant hypothetic scene because we have a quantity of choices concerning such hypotheses–then, perhaps we should instead focus our attention on which scenes may have really occurred in history. If so, it appears more meaningful to wait for the accumulation of evidence and knowledge on prebiotic chemistry and environments–then conducting computer modeling in a conventional way.

It is noteworthy that after we find an optimal parameter-setting in favor of a hypothetic scene, an obviously useful job would be to test the influence of different parameters by altering them separately, especially to see the cases where the model does not work well to support the scene. The detailed information on the optimal parameter-setting and the results of the subsequent testing by separate parameter adjustments would provide clues for us to judge the likelihood of such a scene in history based on our extant knowledge concerning prebiotic chemistry and environments, or for such a judgement in future if by now we have not got relevant knowledge. That is just where the significance of our "reverse" computer modeling lies.

For the automatic parameter-exploration, a large learning rate, though better for a fast learning, may brought about instability in the learning process (Fig 2), especially when the optimum is approaching. In fact, the instability is similar to the common situation for any optimization. An ordinary solution to this problem in machine-learning is to adjust the learning rate progressively, that is, $e$ would decrease gradually when the optimum is approaching. But here such a strategy (or some strategy alike) is not necessary because the aim of parameter-exploration for the evolutionary modeling studies on the origin of life is not to seek an optimum in a pure sense, but to know roughly about the existence of a good parameter set favoring a hypothetic scene. Certainly, for any potential goals to seek an absolute optimum in future work of this area, improvements along this line would be readily to achieve–simply based on those relevant techniques developed in the field of machine-learning [40].

Notably, for the automatic parameter-learning process, the selection of an objective function is a critical step. Different objective functions may result in apparently distinct results. When we chose merely the molecular number of NSR as objective function (Fig 2), the resulting parameter setting may favor the spread of control species as well as that of NSR (Fig 4). When the difference between the NSR and the control species was chosen as the objective function (Fig 5A), the resulting parameter setting may favor the spread of NSR but not the control species (Fig 6). When the average replicator length at step $100 \times 10^4$ was chosen as objective function (Fig 8A), the resulting parameter setting is satisfactory in regard of increasing the replicator length in the evolution (Fig 8B). In contrast, when the average replicator length at step $20 \times 10^4$ was chosen as the objective function (Fig 8C), the resulting parameter setting is obviously "unsatisfactory" (Fig 8D)–because the parameter setting optimizing for the replicator length at an early stage does not necessarily represent a good one favoring long replicators at the final balance.

To balance the effectiveness and the computational costs of the machine-learning, we designed a strategy of progressive learning, in which the parameter values learned from a previous stage are set as the starting point of the learning in a following one (S3 Fig). In fact, such a mechanism involves a notion of "transfer learning" [40]–that is, a parameter set resulting from a learning process may be "lent" and used as the starting point of another learning process provided that the two learning cases are similar or clearly related to each other–thereby speeding the latter learning process. Furthermore, the notion of transfer learning may be quite useful in the whole area of evolutionary modeling concerning the origin of life. As mentioned

already, the origin of life represented a remarkable developmental scenario from simplicity to complexity. For example, the two examples used in the present study are merely dealing with the Darwinian evolution at the molecular level and being relatively simple [19,20]. There have already been quite a few modeling studies tackling Darwinian evolution in a later stage, i.e. at the "proto-cellular" level (e.g., see Ref. [21,24,27,28]), which involves much more parameters. That is to say, on account of intrinsic relationship between earlier and later scenes during the origin of life, probably a portion of parameters in the complex models, i.e. those having been explored in previous simpler models, need not be explored *ab initio*.

In addition to the optimization approach of the gradient ascent, we have shown that the approach of coordinate ascent (see Methods) may also work in the automatic parameter-exploration (S1D Fig), wherein parameters are explored one by one, with iterative cycles. However, in principle of machine-learning, the coordinate ascent is less powerful than the gradient ascent, especially when the parameters are strongly "interdependent". Indeed, here we have observed the interdependence between parameters–during the automatic parameter-adjustment, some parameters increase in their values in certain stages while decreasing in other stages (Figs 3 and S2). This inconsistence of tendency indicates that a "better" value for one parameter may depend on the values of other parameters in use. Therefore, it is expected that, in this area, the approach of the gradient ascent would be more robust than that of coordinate ascent–especially for those complex models concerning the origin of life, in which the interdependence between parameters might be more significant.

Certainly, there may also be other learning algorithms for the automatic parameter-exploration. For example, as a variant of the Metropolis algorithm, we may make small random changes to the parameters, and then accept all moves that improve the objective function and accept moves that decrease the objective function with a smaller probability that depends on the scale of the decrease (accepting decrease to an extant might avoid being trapped permanently in local optima). The idea is no doubt interesting. But the Metropolis algorithm is sometime not quite efficient in finding an optimum. Concerning the automatic parameter-exploration, perhaps further investigation on this method is needed to draw a conclusion.

Another interesting idea is to apply the evolutionary algorithms. For instance, we may treat the investigated parameters as genotypes and the objective function as the phenotype. Genotypes favoring the improvement of the phenotype is selected iteratively over many rounds of "reproduction" where mutation and recombination of the genotypes are introduced. Then, starting from a "bad" parameter-set, perhaps a good parameter-set favoring the hypothetic scene may gradually emerge in the "parameters' evolution". Obviously, the effectiveness and the efficiency (if effective) of the evolutionary algorithms in regard to the automatic parameter-exploration are also expected to be evaluated in future.

No matter how, the automatic parameter-exploration in principle of machine-learning demonstrated in the present study is inspiring. With less scruples regarding the burden of searching for "appropriate" parameters, it is anticipated that researchers in this area would establish models considering more details of prebiotic chemistry (thus with more parameters). Therewith, the results and conclusions derived from the computer simulations would be more comparable with reality and more convincing. Then perhaps the two aspects of scientific efforts concerning the origin of life, experimental and theoretical, would become much more cross-referenced, interdependent, and finally, even merged–thereby hopefully bringing about a breakthrough in the field.

Finally, we notice that the approach we present here may have a broader sense. Although as mentioned above, the reverse way of modeling is particularly important (and effective) in the field of the origin of life, obviously, in other fields of modeling sometimes we might also want to know what a kind of parameter setting would bring about a definite outcome of interest,

thus tending to utilize the reverse way. Then the presented approach of automatic parameter-exploration in principle of machine-learning would be significant.

## Methods

### The gradient ascent

Suppose that $n$ parameters are involved in the parameter-exploration: $p_1, p_2, \ldots \ldots, p_n$, and a definite outcome of the model running (i.e., simulation) using these parameters is $O$, which can be denoted as: $O = Model\ \{p_1, p_2, \ldots \ldots, p_n\}$. This definite outcome, which may represent a hypothetic scene, is adopted as the objective function in our machine-learning, which means we aim to obtain a maximum value of $O$ by adjusting the values of $p_1, p_2, \ldots \ldots, p_n$. Certainly, we may also try to minimize $O$ by modifying the parameters, which are called the "gradient descent". To be concise, we will only describe the optimizing direction of "ascent" here (the same below for the approach of coordinate ascent).

Firstly, we obtain $O[0]$ by running the model with initial parameter values $p_1[0], p_2[0], \ldots \ldots, p_n[0]$, which can be described here as:

$$O[0] = Model\ \{p_1[0],\ p_2[0], \ldots \ldots, p_n[0]\} \tag{1}$$

Then, we begin to test the influence of the parameters on the objective function. We change the value of $p_1$ to $p_1[0]^*(1+e)$, where $e$ denotes the learning-rate ($0 < e < 1$), and obtain:

$$O[0]_1 = Model\ \{p_1[0]*(1+e),\ p_2[0], \ldots \ldots, p_n[0]\} \tag{2}$$

Likewise, we change the values of other parameters and obtain:

$$O[0]_2 = Model\ \{p_1[0],\ p_2[0]*(1+e), \ldots \ldots, p_n[0]\}$$

$$\ldots \ldots$$

$$O[0]_n = Model\ \{p_1[0],\ p_2[0], \ldots \ldots, p_n[0]*(1+e)\}$$

Having done these, we begin to calculate the difference of the objective function brought by the change of parameters:

$$\triangle O[0]_1 = O[0]_1 - O[0] \tag{3}$$

$$\triangle O[0]_2 = O[0]_2 - O[0]$$

$$\ldots \ldots$$

$$\triangle O[0]_n = O[0]_n - O[0]$$

Then the maximal absolute value of these differences is found out:

$$\text{Max\_abs\_} \triangle O[0] = \text{Max}\ \{|\triangle O[0]_1|,\ |\triangle O[0]_2|, \ldots \ldots, |\triangle O[0]_n|\} \tag{4}$$

Based on this, the parameter values for the next learning-round is calculated:

$$p_1[1] = p_1[0]*(1 + e* \triangle O[0]_1 / \text{Max\_abs\_} \triangle O[0]) \tag{5}$$

$$p_2[1] = p_2[0] * (1 + e * \triangle O[0]_2 / \text{Max\_abs\_} \triangle O[0])$$

$$\ldots \ldots$$

$$p_n[1] = p_n[0] * (1 + e * \triangle O[0]_n / \text{Max\_abs\_} \triangle O[0])$$

Subsequently, a new round of learning starts from the step marked by Formula (1) above, and so on, iteratively.

In the approach, the notion of gradient ascent has been manifested in the determination of the parameter values in a new round. For example, if the increase of a parameter ($p_x$) could bring about the greatest improvement regarding the objective function (calculated in the testing stage represented by Formula (2), and judged in Formula (3) and Formula (4)), in the new round (as demonstrated in Formula (5)), it would be adjusted upwards in a full scale, i.e., $p_x = p_x*(1+e)$. In contrast, other parameters, which have less potential to improve the objective function, would be modified in a scale in proportion to its potential. Importantly, in the learning algorithm, the changing directions of the parameter and the objective function are interrelated–for instance, if the increase of a parameter ($p_x$) brings about the decrease of the objective function, in the new round (as calculated in Formula (5)), it would be adjusted downwards–due to the sign of $\triangle O[0]_x$. For details, one can refer to the source code of our learning algorithm (See Code availability). No matter how, the approach here is choosing the steepest direction leading to the maximization of the objective function–in term of mathematics, $\triangle O[0]_x$ represents the changing rate of the objective function $O$ relative to the change of $p_x$, and a vector comprising all these relative changing rates just represents the "gradient". In practice, $\triangle O[0]_x / \text{Max\_abs\_} \triangle O[0]$ is a normalized form of the changing rate $\triangle O[0]_x$.

Notably, here we use the "learning rate" "$e$" in two places, i.e., in Formulas (2) and (5) respectively. However, conceptually, while the one in (5) is the veritable learning rate, the one in (2) actually simply defines the small change of the parameters used to estimate the gradient. For a deterministic model (e.g., the differential equation model as mentioned in introduction), in which the gradient could be determined by analytic methods, the usage of $e$ in sense of that in (2) is often unnecessary. But for the Monte-Carlo model as demonstrated here, we need a "testing rate" to estimate the gradient. Though in principle, the "testing rate" and the true "learning rate" need not be the same, they should adopt a similar scale in practice. For instance, when we want to conduct a fast learning and thus use a great "learning rate" for adjusting the parameters, then using a very small "testing rate" for an accurate estimation of the gradient (which would bring great computational cost) would appear unnecessary; on the other hand, if we would like to conduct a smooth learning and thus use a small learning rate, then using a relatively great "testing rate" would tend to be insufficient to estimate a gradient accurate enough for a valid learning step forward. Therefore, for simplification in practice, here we adopt the same value and use a unitary representation for the two rates.

## Variations on learning strategies

In the approach, the "learning direction" in the testing stage would be adjusted according to the result in the previous round. For example, when the increase of a parameter ($p_x$) brought about the decrease of the objective function in a previous round, in the testing stage of the new round, the parameter would be adjusted downwards instead–that is, in the Formula (2), the actual learning direction might be represented by $p_x[0]*(1-e)$, instead of $p_x[0]*(1+e)$. In the particular situation when the increase (or decrease) of $p_x$ did not result in the alteration of the

objective function, we typically assume an "upward" direction. But when an "downward" direction is assumed, there would be no problem (S1A Fig). Or we may even change the implementation of the testing by using a strategy of "multiplication" rather than that of "addition": when the increase of a parameter ($p_x$) brought about the decrease of the objective function in a previous round, the parameter would be adjusted downwards in a form of $p_x[0]/(1+e)$, instead of $p_x[0]*(1+e)$ (S1B Fig). Furthermore, the approach is also robust when both these two strategies are used simultaneously, that is, the downward adjustment is in a form of $p_x[0]/(1+e)$, and in the particular situation when the increase (or decrease) of $p_x$ did not result in the alteration of the objective function, the downward direction of learning is adopted (S1C Fig).

## The coordinate ascent

In the beginning round of learning, for the first parameter ($p_1$), while the other parameters are fixed, its initial value ($p_1[0]$), three values adjusted upwards ($p_1[0]*(1+e)$, $p_1[0]*(1+e)^2$, $p_1[0]*(1+e)^3$) and three values adjusted downwards ($p_1[0]*(1-e)$, $p_1[0]*(1-e)^2$, $p_1[0]*(1-e)^3$), are tested respectively. In regard of "favoring" the objective function, the most "outstanding" one among the seven values is reserved for testing the other parameters, as well as for serving as the initial value for testing this parameter in the next round (i.e., $p_1[1]$). When all the parameters have been tested, a new round of exploration starts from the first parameter. That is to say, actually, rather than simultaneous exploration as conducted in the approach of the gradient ascent, the parameters are here explored one by one, iteratively–thus, characterized with the notion of the coordinate ascent (S1D Fig).

## Supporting information

**S1 Table. Parameters used in the automatic exploration**
(PDF)

**S1 Fig. Variations on the machine-learning method.** The starting value of parameters are the same as those in Fig 2A: i.e., $PNF = 4\times10^{-3}$, $PNFR = 0.02$, $PND = 1\times10^{-3}$, $PRL = 2\times10^{-5}$, $PBB = 1\times10^{-5}$, $PAT = 0.5$, $PFP = 0.1$, and $PMV = 1\times10^{-3}$. The learning rate $e = 0.5$ (corresponding to the blue line in Fig 2A). As with the cases in Fig 2, the objective function is the number of NSR molecules at step $1.5\times10^5$ in the evolutionary dynamics (refer to Fig 1B). (**a**) When a change of one parameter does not bring about the value change of the objective function, in the next round of learning it is adjusted downwards instead of upwards. (**b**) The change of a parameter concerning the learning rate is implemented by the rule of multiplication, instead of the rule of addition. (**c**) Adoption of the implementation strategy in ***b*** with the adjusting strategy of ***a***. (**d**) Here, instead of the gradient ascent, the approach of coordinate ascent is used. See Methods for a detailed description about all these variations.
(TIF)

**S2 Fig. The automatic parameter-adjustment during the machine-learning–optimizing for longer replicators.** The improvement of replicator length of this case is shown in Fig 8A. The vertical axis of a subfigure represents the value of the corresponding parameter whose name is shown within the panel. The eight parameters are adjusted simultaneously in a learning round.
(TIF)

**S3 Fig. The progressive parameter-exploration for the sake of saving computational costs.** (**a**) The black line is the same as that in Fig 8A, whose objective function, all through the learning process, is the average length of replicators at step $100\times10^4$. The learning process denoted by the magenta segment (from round 1 to 15) uses an objective function of the replicator length at step $10\times10^4$, with a starting point of parameter setting the same as that of the black

line case (refer to Fig 7B); the learning process denoted by the green segment (from round 16 to 30) uses an objective function of the replicator length at step $20\times10^4$, with a starting point of parameter setting achieved from the previous learning process (the magenta segment); the brown segment (from round 31 to 45) uses an objective function of the replicator length at step $40\times10^4$, with a starting point of parameter setting achieved from the previous learning process (the green segment); the red segment (from round 46 to 60) uses an objective function of replicator length at step $100\times10^4$, with a starting point of parameter setting achieved from the previous learning process (the brown segment). The arrows indicate "odd points" (see the legend of Fig 8 for an explanation). (**b**) The black dots represent the improved evolutionary dynamics (in comparison with Fig 7B) by adopting the parameter set at the 60th learning round of the black line case in *a*, which does not use the progressive strategy (actually the same as shown in Fig 8B), while the red dots represent the improved evolutionary dynamics (in comparison with Fig 7B) by adopting the parameter set at the 60th learning round of the red line case in *a*, which uses the progressive strategy.
(TIF)

## Acknowledgments

A portion of the present work are based upon the source code of a computer modeling study from Szathmáry and coworkers [19].

## Author Contributions

**Conceptualization:** Wentao Ma.

**Data curation:** Wentao Ma.

**Funding acquisition:** Wentao Ma.

**Investigation:** Yuzhen Liang, Wentao Ma.

**Methodology:** Chunwu Yu, Wentao Ma.

**Validation:** Wentao Ma.

**Writing – original draft:** Yuzhen Liang, Wentao Ma.

**Writing – review & editing:** Wentao Ma.

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
