## [Decision Letter · Decision Letter 0]

1 Nov 2021

Dear Dr. Ma,

Thank you very much for submitting your manuscript "The automatic parameter-exploration in principle of machine-learning: Powering the evolutionary modeling on the origin of life" for consideration at PLOS Computational Biology. As with all papers reviewed by the journal, your manuscript was reviewed by members of the editorial board and by several independent reviewers. The reviewers appreciated the attention to an important topic. Based on the reviews, we are likely to accept this manuscript for publication, providing that you modify the manuscript according to the review recommendations.

The reviewers have raised various points about the algorithm that should be fully addressed. In particular, many aspects of the algorithm developed and its connection to other algorithms need to be addressed. Added information is needed both on the method development and discussion on its application. Further, please revise the manuscript title for clarity, "in principle of machine-learning" is unclear/ambiguous. 

Sincerely,

Tamar Schlick

Associate Editor

PLOS Computational Biology

Kiran Patil

Deputy Editor

PLOS Computational Biology

[LINK]

The reviewers have raised various points about the algorithm that should be fully addressed. In particular, many aspects of the algorithm developed and its connection to other algorithms need to be addressed. Added information is needed both on the method development and discussion on its application.

Reviewer's Responses to Questions

**Comments to the Authors:**

Reviewer #1: The paper starts from the important point that many models used in evolutionary biology and origin of life studies have many parameters, and that finding suitable values of these parameters is difficult by trial and error. The paper gives a method of automatic learning of parameters which directs the parameters via a gradient ascent method towards optimal values. The model of nucleotide synthetase replication is taken from the author's previous work. This is an interesting model related to the RNA World. The current paper demonstrates that good parameter sets for this model can be found by the automatic learning method. I think this is a useful and interesting paper. In practice, the method requires evaluating the model on many different parameter sets. The practicality of the method will depend on how fast the model can be run on each parameter set.

Philosophical points:

We always have a tendency to want to "prove that our model is right" by finding parameters where our model produces a good outcome in comparison to somebody else's model. If authors select parameters by hand, they could be accused of "cherry picking" parameters that demonstrate a preconceived desired outcome. It seems that this automated method does not get rid of this problem, because there will always be parameters in the learning algorithm itself that can be chosen – such as the outcome function O which is optimized, and details of the rules by which the parameters are adjusted. My point is simply that the automated method does not removed potential human biases.

It is often interesting to present results of a model as a function of changing values of a particular parameter. In this way we can see that good outcomes of a model depend on certain values of the parameter. In other words, it is useful see cases where the model does not work as well as the optimum parameter values.

Technical points:

The parameter e is called the learning rate. But e is used in two different ways in the method. Firstly a parameter is changed from p to p(1+e). The change is Deltap = pe, and the gradient is DeltaO/Deltap. Secondly, the parameters are adjusted by an amount proportional to e (equation 5). In the second sense, e is indeed a learning rate because it determines the rate of change of parameters, but in the first sense, e is not a learning rate. It simply determines the small change Deltap used to estimate the gradient. It is therefore slightly confusing to call e the learning rate on line 336. There is no particular reason why the e used for Deltap has to be the same as the e used for adjusting the parameters .

In a deterministic model we would expect that the estimated gradient DeltaO/Deltap has a well-defined limit dO/dp when e tends to 0, and that the estimated gradient should not depend much on e. In a stochastic model, the outcome will fluctuate, even if the model is run twice with the same parameters. So the gradient will be dominated by statistical error if e is too small. Did the authors ever observe this problem in their simulations? For the results shown in Fig 2, the small e case seems to give quite smooth results, so probably e was not too small.

On the other hand, the large e case appears to be unstable, with big drops in the O function occurring frequently. I would suggest that the reason for this is that the parameters are adjusted by an amount that depends on DeltaO/Max-abs-DeltaO (in equation 5). If the parameters are approaching an optimum, then Max-abs-DeltaO will be small, and all the parameters will be shifted by a big amount, thus moving the model a long way from the optimum. It would probably be better to shift the parameters by an amount proportional simply to DeltaO. Then the algorithm would smoothly approach the optimum.

Another option would be to make small random changes to the parameters, and then accept all moves that increase O and accept moves that decrease O with a smaller probability that depends on DeltaO. This would be a variant of the Metropolis algorithm. This would have the potential advantage of avoiding being trapped permanently in local optima, and also it does not require the calculation of the gradient, which is time-consuming because the model has to be evaluated separately for changes in each parameter.

Reviewer #2: This is a valuable contribution, and may help move the field further. There is only one aspect that I am missing from the discussion. The Authors apply mostly the gradient ascent algorithm, but there are several others. Do they have a comment what they would expect (or not) from alternatives? Evolutionary algorithms naturally come to mind in this context. At least some of the investigated parameters could be made genotypic properties and thus evolvable. Parameters that are simply given by chemistry in the given environment (such as probability of breaking a phosphodiester bond) cannot be treated in this way, but others can. I think this would be interesting. It would be a bit similar to population genetic models that have evolvable mutation or recombination rates.

**Have the authors made all data and (if applicable) computational code underlying the findings in their manuscript fully available?**

Reviewer #1: Yes

Reviewer #2: Yes

PLOS authors have the option to publish the peer review history of their article (what does this mean?). If published, this will include your full peer review and any attached files.

Reviewer #1: No

Reviewer #2: **Yes: **Eors Szathmary

Figure Files:

Data Requirements:

Reproducibility:

References:

---

## [Decision Letter · Decision Letter 1]

15 Dec 2021

Dear Dr. Ma,

We are pleased to inform you that your manuscript 'The automatic parameter-exploration with a machine-learning-like approach: Powering the evolutionary modeling on the origin of life' has been provisionally accepted for publication in PLOS Computational Biology.

Best regards,

Tamar Schlick

Associate Editor

PLOS Computational Biology

Kiran Patil

Deputy Editor

PLOS Computational Biology

Reviewer's Responses to Questions

**Comments to the Authors:**

Reviewer #1: The responses to the previous questions are clear. I do not have any further questions.

Reviewer #2: The paper has been duly revised.

**Have the authors made all data and (if applicable) computational code underlying the findings in their manuscript fully available?**

Reviewer #1: Yes

Reviewer #2: Yes

PLOS authors have the option to publish the peer review history of their article (what does this mean?). If published, this will include your full peer review and any attached files.

Reviewer #1: No

Reviewer #2: **Yes: **Eörs Szathmáry

---

## [Editor Report · Acceptance letter]

21 Dec 2021

PCOMPBIOL-D-21-01741R1 

The automatic parameter-exploration with a machine-learning-like approach: Powering the evolutionary modeling on the origin of life

Dear Dr Ma,

I am pleased to inform you that your manuscript has been formally accepted for publication in PLOS Computational Biology. Your manuscript is now with our production department and you will be notified of the publication date in due course.

With kind regards,

Zsofia Freund
